# The combination of quercetin and leucine synergistically improves grip strength by attenuating muscle atrophy by multiple mechanisms in mice exposed to cisplatin

Te-Hsing Hsu[1], Ting-Jian Wu[2], Yu-An Tai[1], Chin-Shiu Huang[3], Jiunn-Wang Liao[4], Shu-Lan Yeh[1,5]*

1 Department of Nutritional Science, Chung Shan Medical University, Taichung, Taiwan, 2 Institute of medicine, Chung Shan Medical University, Taichung, Taiwan, 3 Department of Health and Nutrition Biotechnology, Asia University, Taichung, Taiwan, 4 Graduate Institute of Veterinary Pathology, College of Veterinary Medicine, National Chung Hsing University, Taichung, Taiwan, 5 Department of Nutrition, Chung Shan Medical University Hospital, Taichung, Taiwan

* suzyyeh@csmu.edu.tw

**Data Availability Statement:** All relevant data are within the paper.

## Abstract

Both quercetin and leucine have been shown to exert moderately beneficial effects in preventing muscle atrophy induced by cancers or chemotherapy. However, the combined effects of quercetin and leucine, as well as the possible underlying mechanisms against cisplatin (CDDP)-induced muscle atrophy and cancer-related fatigue (CRF) remain unclear. To investigate the issues, male BALB/c mice were randomly assigned to the following groups for 9 weeks: Control, CDDP (3 mg/kg/week), CDDP+Q (quercetin 200 mg/kg/day administrated by gavage), CDDP+LL (a diet containing 0.8% leucine), CDDP+Q+LL, CDDP+HL (a diet containing 1.6% leucine), and CDDP+Q+HL. The results showed that quercetin in combination with LL or HL synergistically or additively attenuated CDDP-induced decreases in maximum grip strength, fat and muscle mass, muscle fiber size and MyHC level in muscle tissues. However, the combined effects on locomotor activity were less than additive. The combined treatments decreased the activation of the Akt/FoxO1/atrogin-1/MuRF1 signaling pathway (associated with muscle protein degradation), increased the activation of the mTOR and E2F-1 signaling pathways (associated with muscle protein synthesis and cell cycle/growth, respectively). The combined effects on signaling molecules present in muscle tissues were only additive or less. In addition, only Q+HL significantly increased glycogen levels compared to the CDDP group, while the combined treatments considerably decreased CDDP-induced proinflammatory cytokine and MCP-1 levels in the triceps muscle. Using tumor-bearing mice, we demonstrated that the combined treatments did not decrease the anticancer effect of CDDP. In conclusion, this study suggests that the combination of quercetin and leucine enhanced the suppressed effects on CDDP-induced muscle weakness and CRF through downregulating muscle atrophy and upregulating the glycogen level in muscle tissues without compromising the anticancer effect of CDDP. Multiple mechanisms, including regulation of several signaling pathways and decrease in proinflammatory

**Funding:** This research was supported by grants (MOST 108-2320-B-040-014-MY2) from the National Science and Technology Council, Republic of China, as the authors have claimed in the Funding Statement. The funder had no role in study design, data collection and analysis, the decision to publish or preparation of the manuscript.

**Competing interests:** The authors have declared that no competing interests exist.

mediator levels in muscles may contributed to the enhanced protective effect of the combined treatments on muscle atrophy.

## Introduction

Cisplatin (cis-diamine-dichloro platinum (II); CDDP), an inorganic platinum-based chemotherapeutic agent, is used to treat several types of solid tumors such as lung cancer despite the non-specific target cell toxicity [1]. Muscle weakness and cancer-related fatigue (CRF) are common adverse effects induced by CDDP [2,3]. These adverse effects are induced by multi-biological mechanisms including skeletal muscle mass wasting and proinflammatory cytokines upregulation [4]. Dysregulation of protein degradation and protein synthesis contribute to CDDP-induced muscle wasting [3,5]. Since muscle wasting and fatigue increase morbidity and mortality as well as decrease the tolerance to therapy in cancer patients, it is important to determine the strategy for attenuating the above-indicated adverse effects induced by CDDP.

Quercetin is a flavonoid that has antioxidative, antiinflammatory, and anticancer properties via various mechanisms including regulating signaling molecules [6]. It is commonly found in many plant foods and herbs [7]. Quercetin has been shown to suppress chemotherapy-induced adverse effects with enhancing or no effect on the antitumor efficacy [8,9]. Our previous study showed that quercetin given through a daily diet supplemented with quercetin (1%) or i.p. injection (10 mg/kg, 3 times/week) significantly increased the muscle mass in tumor-bearing nude mice exposed to trichostatin A (TSA), accompanied with the enhancing anticancer effect of TSA by significant downregulation of forkhead box O1 (FoxO1) as well as the downstream two muscle-specific ubiquitin ligases, atrogin-1and muscle ring-finger-1 (MuRF-1) [8].

Branched-chain amino acids (BCAAs), especially leucine, are known to control skeletal muscle protein metabolism mainly by stimulating protein synthesis [10]. In tumor-bearing rats, a study showed that a diet containing 3% leucine decreases the loss of lean body mass, gastrocnemius muscle, and myosin content as compared with an isonitrogenous and isocaloric control diet [11]. A few studies also demonstrated that leucine is the key factor that affects muscle protein anabolic responses in rats [12] and older women [13]. The mechanisms underlying the effects of leucine on protein synthesis are associated with the activation of the mammalian target of rapamycin (mTOR) signaling pathway [14]. mTOR is a member of the family of phosphoinositide (PI)3-kinases (PI3K)/Akt, and its activation leads to phosphorylation of its downstream target protein like eukaryotic translation initiation factor 4E-binding protein 1 (4E-BP1) and p70 ribosomal S6 kinase (S6K), which in turn results in the translation of proteins involved in the protein synthesis [10].

A previous study [15] showed an additive effect of a multitargeted approach (combination of fish oil, high protein, and leucine) for improving muscle function and daily activity in tumor-bearing cachectic mice. The authors suggested that through the addition of fish oil, the combined supplement reduces inflammation and catabolism; whereas anabolism is targeted by the presentation of high protein and leucine. Therefore, we hypothesized that the combination of quercetin and leucine could also exhibit a multitargeted approach, including regulation of the E2F-1 signaling pathway, which plays a crucial role in the control of the cell cycle and cell growth [16], in skeletal muscle to attenuate cisplatin-induced muscle wasting and fatigue. Hence, we used CDDP-exposed BALB/c mice to investigate the hypothesis. We also investigated whether the combined supplementation compromised the anticancer effects of CDDP in tumor-bearing mice.

## Materials and methods

### Reagents

Quercetin was purchased from Alfa Aesar (Tewksbury, MA, USA). CDDP was purchased from Acros Organics (Geel, Belgium). AIN-93M with or without leucine supplementation (0.8 or 1.6%) was purchased from TestDiet (Richmond, Indiana, USA). All the other chemicals and reagents used in this study are of analytical grade.

### Animal study

To investigate the issues mentioned above, two animal studies (in BALB/c mice or nude mice) were conducted. Animal care followed the International Guiding Principles for Biomedical Research Involving Animals. The animals were sacrificed with $CO_2$ asphyxiation after 8 or 9 weeks of CDDP treatments and all efforts were made to minimize suffering. All study protocols were approved by the Institutional Animal Care and Use Committee of Chung Shan Medical University (IACUC approval no. 2189); and researchers in this study have completed the Animal Care and Use Training Program provided by the Animal Care and Use Center of Chung Shan Medical University or National Chung Hsing University. First, male BALB/c mice aged 4 weeks were obtained from the National Laboratory Animal Center (Taipei, Taiwan) and were housed in an animal room with an alternating 12-h light/dark cycle, controlled temperature (25 ˚C) and humidity (50–60%) [9]. After being acclimated for 1 week, the animals were randomly assigned to the following seven groups (n = 8/group) for 9 weeks: Control, CDDP, CDDP+Q, CDDP+LL, CDDP+Q+LL, CDDP+HL, and CDDP+Q+HL. CDDP was administered intraperitoneally (i.p) at a dose of 3 mg/kg body weight per week [3,9] by dissolving in a 0.9% saline solution. Quercetin (Q) was dissolved in a 0.05% xanthan gum solution and administered at a dose of 200 mg/kg body weight daily by oral gavage; while leucine was administered by a diet containing 0.8% (low dose, LL) or 1.6% (high dose, HL) leucine. All animals were checked daily to assess general animal health and behavior. As mentioned above, the mice were sacrificed by $CO_2$ asphyxiation after nine weeks of treatment. The doses of quercetin and leucine were chosen according to our previous study [8] and the study of Norren et al. [15], respectively. The control group served as the vehicle. All animals were allowed free access to the AIN-93M diet (a commercial diet containing protein, fat, and carbohydrates: 13.7%, 9.8%, and 76.5%, respectively) or a leucine-supplemented diet (AIN-93M diet supplemented with 0.8% or 1.6% leucine) and water during the study. The protein level of the diet supplemented with 0.8% or 1.6% leucine increased to 14.3% and 14.9%, respectively, while the carbohydrate levels decreased to 75.9% and 75.3%, respectively. The calories (kcal/g) of all 3 diets were similar, approximately 3.75 kcal/g. Body weight and food intake of the animals were recorded weekly during the experiment. Furthermore, the maximum grip strength and physical activity of the mice were determined after CDDP injection for 24 hours at weeks 7 and 8, respectively. After being sacrificed, blood samples were collected and the plasma sample was separated and used for various biochemical analyses. Meanwhile, heart, liver, spleen, kidney, testes, epididymal fat, as well as parts of muscles, including triceps, gastrocnemius, soleus, and tibialis anterior, were collected and stored at -80 ˚C until analysis. The quadriceps muscles were stored in 10% formalin and then fixed and sectioned for H&E staining to determine the fiber size.

Furthermore, xenograft tumor model mice were used to investigate whether quercetin and leucine alone or in combination compromised the anticancer effect of CDDP. Male nude mice (aged 4 weeks) were housed in specific pathogen-free cages with the conditions mentioned above. After being acclimated for 1 week, the mice were injected with A549 cells (a human

lung cancer cell line) into the flank as described previously [9]. Four weeks after tumor cell injection, the tumor-bearing mice were randomly assigned to the following groups (n = 5/group) for 8 weeks: Control, CDDP, CDDP+Q, CDDP+HL, and CDDP+Q+HL. CDDP was administered at a dose of 5 mg/kg body weight per week (i.p), while Q and HL were given as mentioned above. We chose a higher dose of CDDP in nude mice based on the previous study [9]. All animals were also checked daily to assess general animal health and behavior. After the experiment, the animals were also sacrificed by asphyxiation with $CO_2$ to determine the epididymal fat and muscle weight.

## Forelimb grip strength test

The BALB/c mice were subjected to the forelimb grip strength test to measure the maximum grip strength (MGS) using a grip strength meter (Ugo Basile, Italy) after CDDP injection at week 7 (all groups). The mice were allowed to grip the triangle bar with two forelimbs and then the mouse's tail was gently pulled back. The maximum force was recorded when the mice released the grasp. The tests were performed three times at one-min intervals by the same person using a similar and stable force for each mouse to obtain the average value. The maximum force value was used to reflect muscle weakness.

## Locomotor activity

At week 8, four mice from each group were individually housed in transparent cages (17 x 28 x 12.5 cm) after CDDP injection, and 12 hours later locomotor activities were monitored and recorded (from 11:00 pm to 01:00 am). We chose this time interval because the mice were active during that particular time. Then, movement and rest were analyzed using the Video Trace Mouse II software (SINGA, Taiwan).

## Western blotting

Gastrocnemius muscle tissues (0.03 g) were homogenized in 300 μL RIPA buffer (150 mM sodium chloride, 50 mM Tris-HCl, 1% nonidet P-40 (NP-40), 0.5% sodium deoxycholate, and 0.1% SDS) to determine protein levels of atrogin-1 (Cat #: AP2041, ECM biosciences, Versailles), MuRF1 (Cat #: MP3401, ECM biosciences), p-AktSer473 (Cat #: 4060, Cell Signaling Technology)/Akt (Cat #: 4691, Cell Signaling Technology), p-FoxO1Thr24 (Cat #: 9464, Cell Signaling Technology)/FoxO1 (Cat #: 9454, Cell Signaling Technology), p-mTORSer2448 (Cat #: 5536, Cell Signaling Technology)/mTOR (Cat #: 2983, Cell Signaling Technology), p-p70S6KThr389 (Cat #: 9234, Cell Signaling Technology)/p70S6K (Cat #: 9202, Cell Signaling Technology), p-4E-BP1Thr37/46 (Cat #: 2855, Cell Signaling Technology)/4E-BP1 (Cat #: 9644, Cell Signaling Technology), p-RBSer807/811 (Cat #: 8516, Cell Signaling Technology)/RB (Cat #: 9309, Cell Signaling Technology), proliferating cell nuclear antigen (PCNA, cat #: ab29, Abcam), type IIa myosin heavy chain (MyHC, cat #: sc-53095, Santa Cruz Biotechnology), E2F-1 (Cat #: sc-251, Santa Cruz Biotechnology), Cyclin D (Cat #:sc-8396, Santa Cruz Biotechnology), CDK4 (Cat #: sc-23896, Santa Cruz Biotechnology), and GAPDH (Cat #: sc-47724, Santa Cruz Biotechnology) in muscle tissues by western blotting as described in detail previously [8].

## Muscle fiber size

The quadriceps of mice were fixed and embedded in a cassette and immersed in formalin before being sliced and followed by H&E staining [17]. Then, quadriceps samples were examined using Tissue Cytometer (TissueGnostics, Vienna, Australia; magnification, x200).

Myofiber CSAs of the rectus femoris region were calculated using TissueFAXS Viewer software (TissueGnostics, Vienna, Australia), and a minimum of 8 random images and 200 sets (25 sets/image) of data were acquired per group.

## Glycogen, monocyte chemoattractant protein-1 (MCP-1), and proinflammatory cytokine levels

The levels of glycogen in the triceps muscle tissue were determined using a Glycogen Colorimetric/Fluorometric Assay kit (BioVision, USA). MCP-1 and proinflammatory cytokine (TNF-α, IL-6, and IL-1β) concentrations in the triceps muscle tissue were determined using an enzyme-linked immunosorbent assay (ELISA) kit (R&D Systems, Minneapolis, MN, USA; Thermo Fisher, Waltham, MA, USA). The triceps muscle tissue samples were prepared as described previously [8]. We determined the glycogen and cytokine levels of the triceps muscle because this muscle is the muscle located in the forelimb, which was used to measure the MGS.

## Statistical analysis

All data are expressed as mean ± standard deviation (SD). Statistical analysis was performed using SPSS software, Ver. 18 (IBM, USA). Group differences were assessed by one-way analysis of variance (ANOVA) followed by Tukey HSD test for comparison of group means or independent sample t-test for two-group comparisons. In addition, we determined whether the combined preventive effects of quercetin and leucine on CDDP-induced damages were synergistic or not using a one-sample t-test. When the observed inhibition of the combined treatment for one parameter was significantly better than its expected inhibition, a synergistic effect was evident; when the difference between the observed and expected inhibitions was not significant, an additive effect was indicated. The observed inhibition (%) of one parameter was calculated as (|the value of treatment–the value of CDDP|/ the value of CDDP) × 100%; the expected inhibition was calculated as the observed inhibition of quercetin + the observed inhibition of leucine (LL/HL). P values < 0.05 are considered statistically significant.

## Results

### Body weight and food intake

As shown in Fig 1A, the mean body weight of BALB/c mice before intervention (week 1) did not show any significant difference between the groups. However, after 9 weeks of treatment, the mean body weight of the CDDP group was lower than that of the control group by 37% (p < 0.05), while LL, or HL alone or in combination with Q significantly attenuated CDDP-induced decrease in body weight in the order Q+HL, Q+LL, HL, LL, Q. The mean body weight in the Q+HL group was significantly higher than in the Q or HL alone groups. In addition, we found that compared to the control group, the mean intake of mice exposed to CDDP alone was significantly reduced from week 4. Supplementation with LL or HL alone or in combination with Q significantly attenuated the adverse effect of CDDP at week 9 (Fig 1B). The combined supplements had an additive effect on body weight and food intake (S1 Table in S1 File).

### Grip strength and locomotor activity

We determined the MGS and locomotor activity as markers of muscle weakness and CRF in mice exposed to CDDP. The results showed that mice exposed to CDDP had a significant decrease of 35% in MGS compared to the control group (Fig 2A). In contrast, MGS increased significantly in the HL, but not in the Q and LL alone group compared to the CDDP alone group. The enhanced effects on MGS were significant and synergistic when LL or HL was

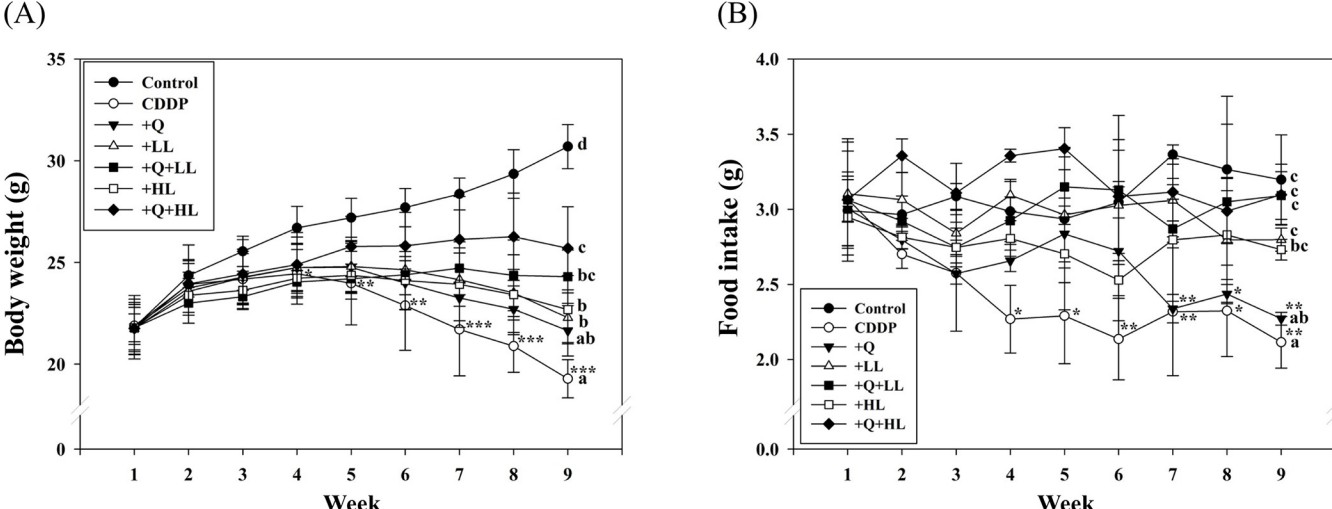

**Fig 1.** The individual and combined effect of quercetin (Q) and low dose (LL) or high dose (HL) of leucine on body weight (A) and food intake (B) in BALB/c mice exposed to cisplatin (CDDP). Values (mean ± SD) not sharing common letters are significantly different (one-way ANOVA, $p < 0.05$). Whereas a *, **, and *** denote a significant difference from the control group (Student's t-test, $p < 0.05$, $p < 0.01$, and $p < 0.001$, respectively).

combined with Q (S2 Table in S1 File). Furthermore, a similar trend was found in the locomotor activity test, which shows that mice barely moved 12 hours after CDDP injection (Fig 2B). All mice in the treatment groups showed a significant improvement in activity in the following order: Q+HL, Q+LL, HL, LL, Q. The combined effects of Q and HL or LL on locomotor activity were less than additive (S2 Table in S1 File).

## Fat and muscle weight

As we have observed previously [9], CDDP significantly decreased epididymal fat mass (Fig 3A). In addition, CDDP significantly decreased total muscle weight by about 26% (Fig 3B)

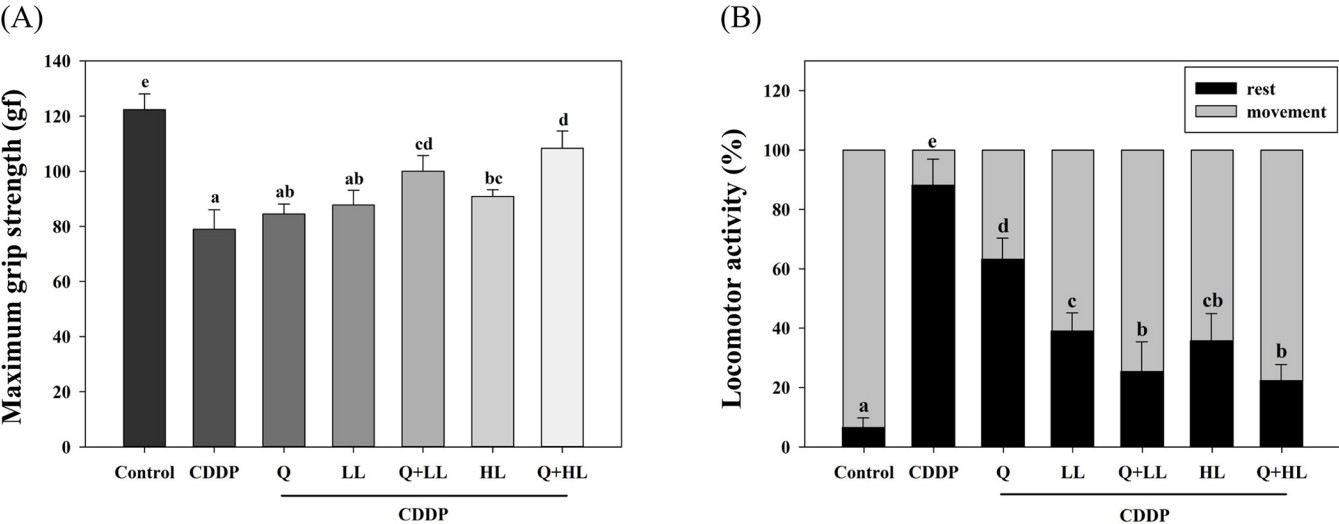

**Fig 2.** The individual and combined effect of quercetin (Q) and low dose (LL) or high dose (HL) of leucine on maximum grip strength (A) and locomotor activity (B) in BALB/c mice exposed to cisplatin (CDDP). The maximum grip strength was recorded at 24 hours after CDDP injection at week 7, and locomotor activity was recorded 12 hours after CDDP injection at week 8. Values (mean ± SD) not sharing common letters are significantly different (one-way ANOVA, $p < 0.05$).

(A)

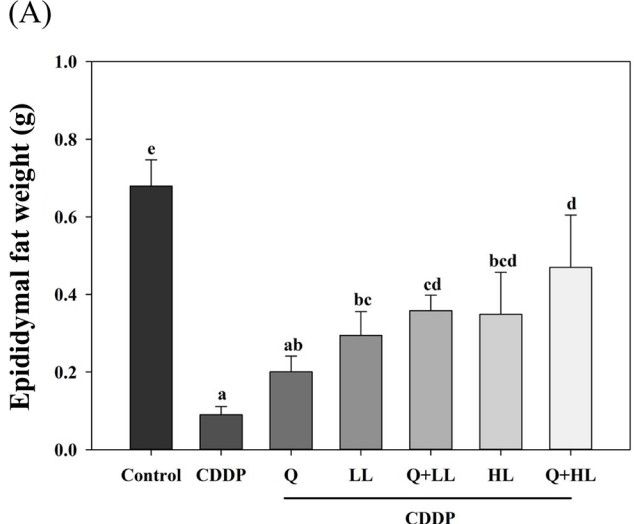

(B)

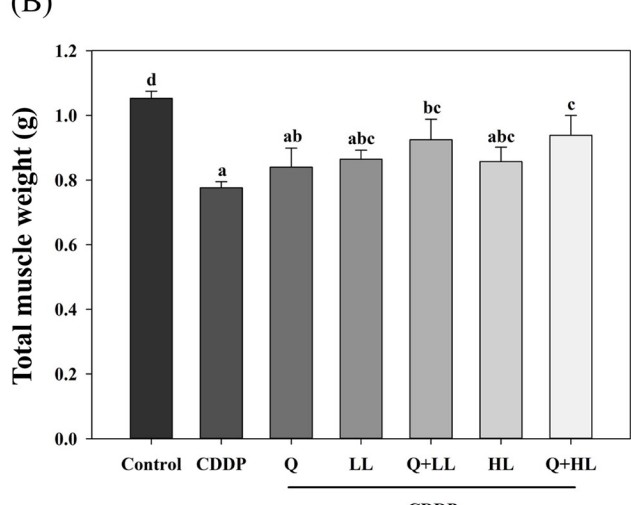

**Fig 3.** The individual and combined effect of quercetin (Q) and low dose (LL) or high dose (HL) of leucine on epididymal fat (A) and total muscle (B) weight in BALB/c mice exposed to cisplatin (CDDP). Values (mean ± SD) not sharing common letters are significantly different (one-way ANOVA, $p < 0.05$).

and also the individual weight of the triceps, quadriceps, gastrocnemius, soleus, and tibialis anterior(Table 1). Q in combination with LL or HL (especially Q+HL) significantly improved the weight of fat, total muscle, triceps, quadriceps, gastrocnemius, and soleus mass. In general, the protective effects of various supplements were in order as found in MGS. However, the combination of Q and LL or HL only had an additive effect on fat and muscle weight (S3 and S4 Tables in S1 File).

## Myofiber size

Furthermore, we performed a morphometric analysis of the quadriceps muscle, a fast-twitch myofiber-dominant muscle, using H&E stain and measured the muscle fiber size (Fig 4A–4C). The H&E stained muscle slides of CDDP-exposed mice showed muscle atrophy; CDDP significantly reduced the mean CSA of quadriceps myofibers by 62%. Q, LL, and HL alone or combined supplements significantly attenuated the effect of CDDP; the combined effects of quercetin and leucine were significantly better than the individual effects. However, only the combined effect of Q and HL was synergistic (S5 Table in S1 File, $p = 0.024$). The highest

**Table 1. The individual and combined effect of quercetin (Q), and low dose (LL) or high dose (HL) of leucine on various muscle weights in BALB/c mice exposed to cisplatin (CDDP).**

|  | Triceps# | Quadriceps | Gastrocnemius | Soleus | Tibialis anterior |
|---|---|---|---|---|---|
|  | (mg) | | | | |
| Control | 254 ± 09[c] | 426 ± 22[d] | 295 ± 06[d] | 14.0 ± 0.7[d] | 81.2 ± 2.6[b] |
| CDDP | 179 ± 08[a] | 304 ± 12[a] | 209 ± 09[a] | 10.4 ± 0.9[a] | 60.4 ± 1.3[a] |
| CDDP+Q | 194 ± 17[ab] | 344 ± 24[ab] | 228 ± 17[ab] | 11.2 ± 0.8[ab] | 64.2 ± 2.9[a] |
| CDDP+LL | 200 ± 02[ab] | 353 ± 13[bc] | 235 ± 11[bc] | 11.4 ± 1.1[ab] | 64.8 ± 4.0[a] |
| CDDP+Q+LL | 206 ± 16[b] | 378 ± 32[bc] | 250 ± 18[bc] | 11.4 ± 1.1[ab] | 67.8 ± 7.8[a] |
| CDDP+HL | 196 ± 15[ab] | 348 ± 23[abc] | 238 ± 09[bc] | 12.2 ± 0.4[bc] | 65.2 ± 1.8[a] |
| CDDP+Q+HL | 214 ± 18[b] | 393 ± 24[cd] | 254 ± 13[c] | 13.8 ± 0.8[cd] | 67.8 ± 3.8[a] |

#Values (mean ± SD) not sharing common letters are significantly different (one-way ANOVA, $p < 0.05$).

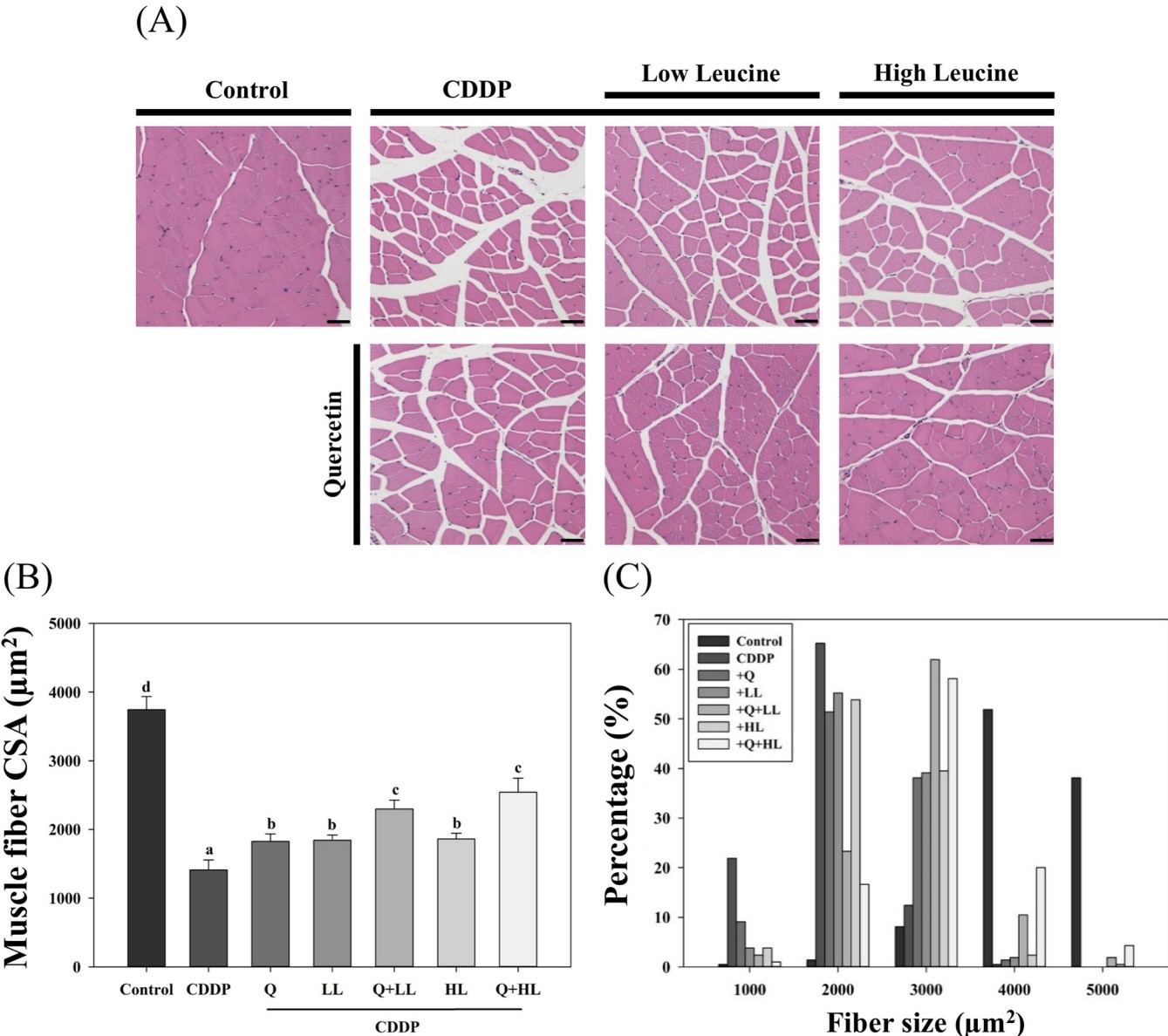

**Fig 4.** The individual and combined effect of quercetin (Q) and low dose (LL) or high dose (HL) of leucine on H&E staining images with scale bar 50 μm (A), mean cross-sectional area (CSA) of muscle fiber (B), and fiber size distribution (in percentage; C) in the quadriceps muscle. Values (mean ± SD) not sharing common letters are significantly different (one-way ANOVA, $p < 0.05$).

proportions of CSA in the Q+LL and Q+HL groups were at 3000 μm$^2$ (vs. 2000 μm$^2$ in the CDDP group), and the mean CSA in these two groups increased by 63% and 80%, respectively.

## FoxO1 signaling pathway in muscle tissues

We also determined the protein expression of the MyHC (type IIa), which is a type of fast muscle fiber and a common skeletal muscle protein content index [18], in the gastrocnemius (another fast-twitch myofiber-dominant muscle) to confirm the finding above. Consistently, the results showed that MyHC was significantly reduced by 44% after exposure to CDDP compared to the control group (Fig 5A). HL alone and two combined treatments significantly

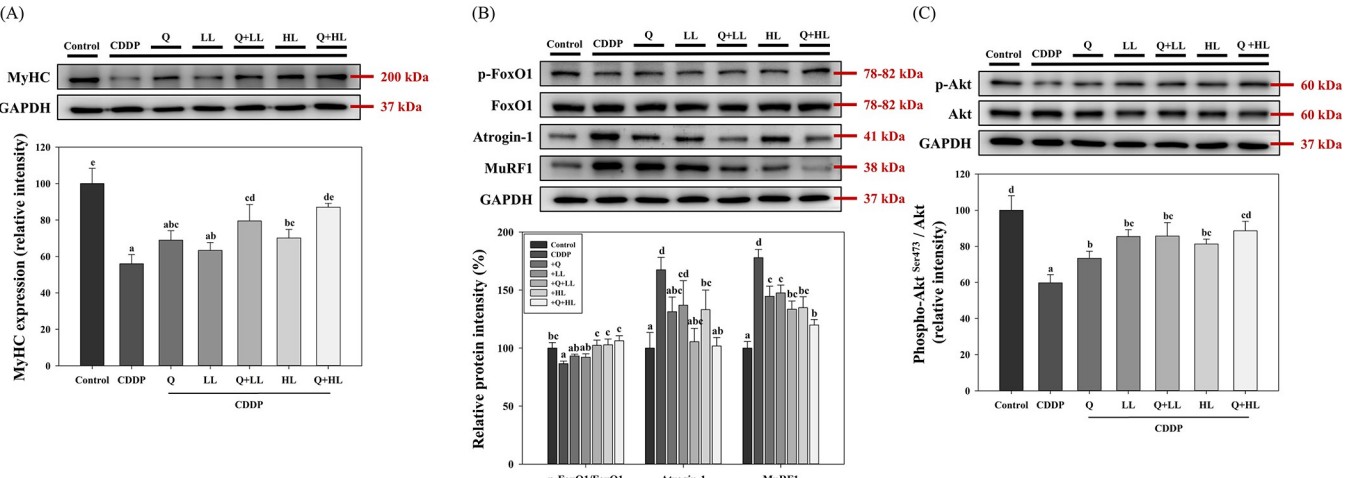

**Fig 5.** The individual and combined effect of quercetin (Q) and low dose (LL) or high dose (HL) of leucine on relative protein expression of MyHC (A), p-FoxO1/FoxO1, Atrogin-1, and MuRF1 (B) as well as p-Akt/Akt (C) in the gastrocnemius muscle in BALB/c mice exposed to cisplatin (CDDP). Values (mean ± SD) not sharing common letters are significantly different (one-way ANOVA, $p < 0.05$).

attenuated the decrease in MyHC expression induced by CDDP. The combination of Q with HL significantly and synergistically (S6 Table in S1 File) enhanced the recovery of MyHC levels compared with the individual supplement. To examine the regulation of protein degradation by Q, LL, or HL alone or in combination in muscle exposed to CDDP, we then evaluated the phosphorylation of FoxO1(inactive form), a transcription factor, as well as the protein expression of the downstream targets Atrogin-1 and MuRF1, which are muscle atrophy-related ubiquitin ligases [5] in the gastrocnemius. As shown in Fig 5B, CDDP significantly decreased the ratio of p-FoxO1/ FoxO1, indicating CDDP increases the activity of FoxO1 signaling. Also, CDDP significantly increased the protein expression of Atrogin-1 and MuRF1 compared with the control group. Q, LL, and HL alone or in combination tended to attenuate all the effects of CDDP, however, only the effects of HL, Q+LL, and Q+HL were significant in all parameters. Because FoxO1 is a downstream target of Akt, a serine-threonine protein kinase, we also determined the ratio of p-Akt/Akt. The results showed that the trend for activation (phosphorylation) of Akt was consistent with the phosphorylation of FoxO1 (Fig 5C). The combined effects of Q and LL/HL on the Akt/FoxO1/MuRF1/Atrogin-1 signaling pathway were not synergistic (S6 Table in S1 File).

## mTOR signaling pathway in muscle tissues

In addition, we determined the expression of the mTOR signaling pathway, which is the main anabolic pathway that regulates protein synthesis in skeletal muscle [19]. As seen in Fig 6, CDDP significantly decreased the phosphorylation of mTOR, p70 S6K, and 4E-BP1 by 28%, 13%, and 23%, respectively compared with the control group. HL and LL alone or in combination with Q significantly increased the phosphorylation of mTOR, p70 S6K, and 4E-BP1. The presence of Q only significantly enhanced the effects of LL or HL on the ratio of p-4E-BP1/ 4E-BP1 in mice exposed to CDDP; however, the combined effect of Q and LL/HL were still not synergistic (S7 Table in S1 File).

## E2F-1 signaling pathway in muscle tissues

To investigate whether the regulation of cell cycle or cell growth is also involved in the increases in muscle mass by various supplements, we also examined the expression of the E2F-

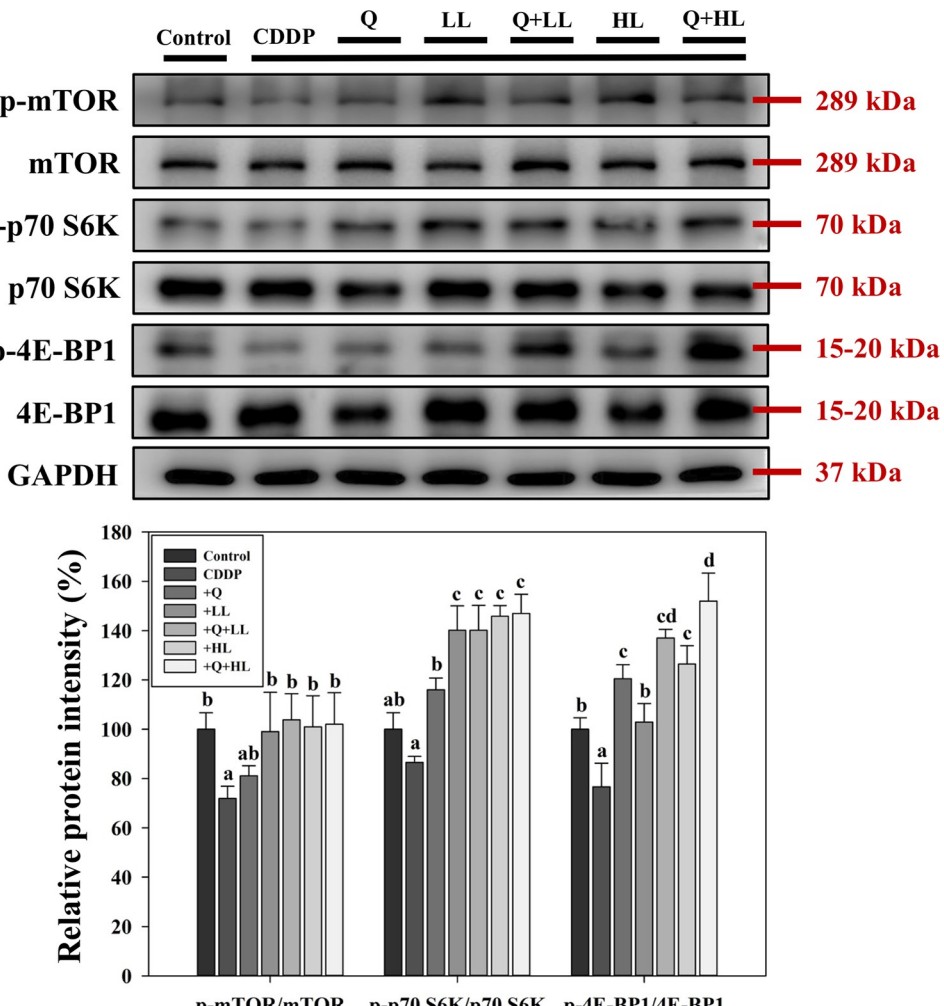

**Fig 6. The individual and combined effect of quercetin (Q) and low dose (LL) or high dose (HL) of leucine on relative protein phosphorylation of mTOR, p70 S6K, and 4E-BP1 in the gastrocnemius muscle in BALB/c mice exposed to cisplatin (CDDP).** Values (mean ± SD) not sharing common letters are significantly different (one-way ANOVA, p < 0.05).

1 signaling pathway in skeletal muscle. CDDP consistently tended to decrease the phosphorylation of RB, as well as the expression of E2F-1, Cyclin D, and CDK4 protein by 21–32%. (Fig 7A). However, the effect of CDDP on E2F-1 was not significant. Except for E2F-1, Q+HL and Q+LL tended to recover the changes in p-RB/RB, Cyclin D, and CDK4 levels induced by CDDP better than or similar to that of the single compound. Consistently, LL, or HL alone or in combination with Q significantly and similarly increased the expression of PCNA protein, a marker of cell proliferation, in muscle tissues of mice exposed to CDDP (Fig 7B). All of the combined effects of quercetin and leucine were additive or less than additive (S8 Table in S1 File).

## Glycogen and proinflammatory cytokines

It has been well documented that the glycogen stored in muscle could affect the strength of the muscle [20]. Because the mice were subjected to the forelimb grip strength test to measure the MGS, we, therefore, analyzed the glycogen and proinflammatory cytokine levels of the triceps

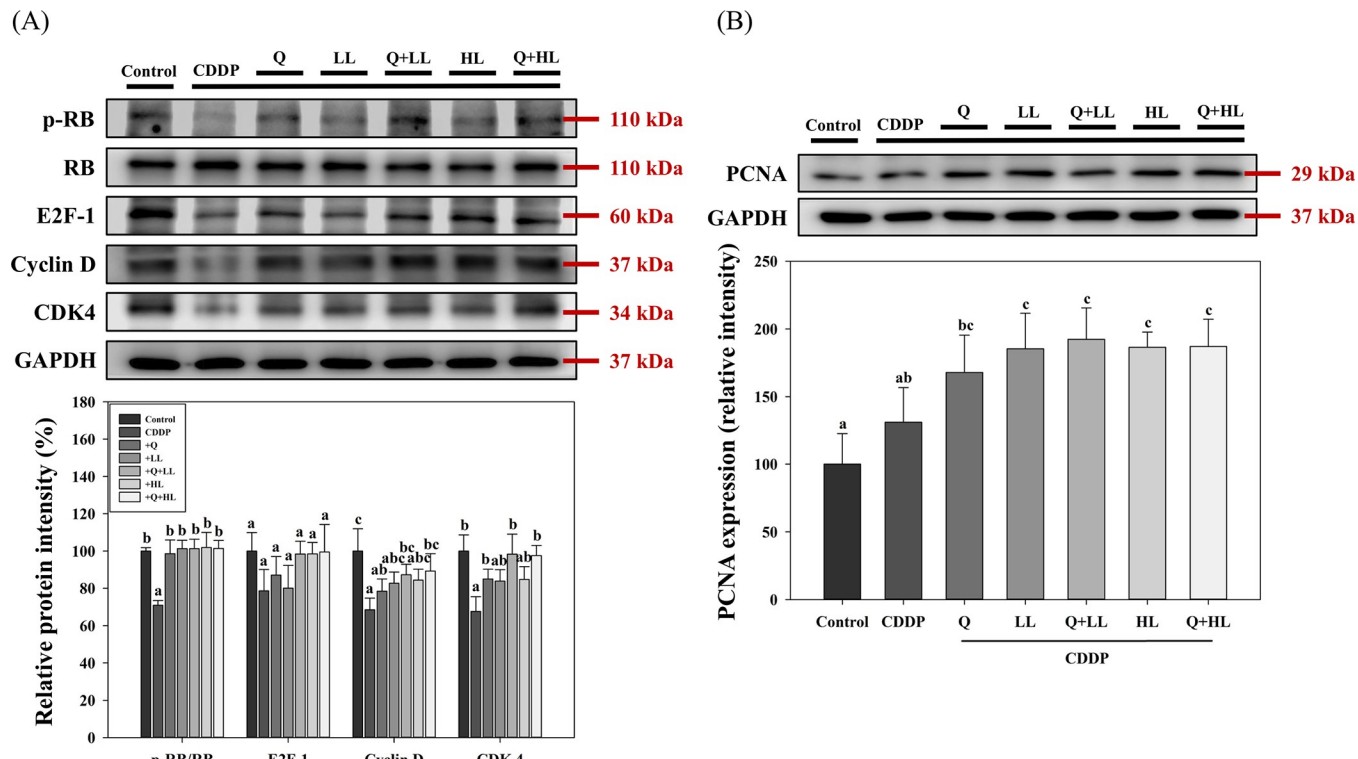

**Fig 7.** The individual and combined effect of quercetin (Q) and low dose (LL) or high dose (HL) of leucine on relative protein expression of p-RB, E2F-1, Cyclin D, CDK4 (A) and PCNA (B) in the gastrocnemius muscle in BALB/c mice exposed to cisplatin (CDDP). Values (mean ± SD) not sharing common letters are significantly different (one-way ANOVA, $p < 0.05$).

muscle, which is the muscle located in the forelimb. As shown in Fig 8A, the glycogen level in the CDDP group was decreased considerably by 42% compared to the control group. All supplements tended to recover the glycogen level but only Q+HL significantly increased glycogen levels in the triceps muscle compared with the CDDP alone group. In addition, CDDP increased the levels of proinflammatory mediators, including MCP-1, TNF-α, IL-6, and IL-1β in the triceps muscles (Fig 8B–8E). All supplemented groups significantly decreased CDDP-induced proinflammatory cytokine levels in muscle in the following order: Q+HL, Q+LL > Q, HL, LL. The combined treatments did not show synergistic effects on decreasing the levels of glycogen and proinflammatory cytokines (S9 Table in S1 File).

## Tumor growth in tumor-bearing nude mice

Using a nude mouse xenograft model, we further investigated the effect of quercetin in combination with HL on the anticancer effect of CDDP. The results showed CDDP alone or in combination with various supplements did not significantly affect the food intake (Fig 9A). Q and HL alone or combined also did not significantly affect the suppressed effects of CDDP (5 mg/kg) on tumor size and body weight (Fig 9B and 9C). However, the combination of Q and HL consistently and significantly decreased CDDP-induced epididymal fat and gastrocnemius muscle weight loss (Fig 9D and 9E). The combined treatment revealed an additive protective effect on CDDP-induced fat and muscle weight loss (S10 Table in S1 File) without hampering the anticancer activity of CDDP.

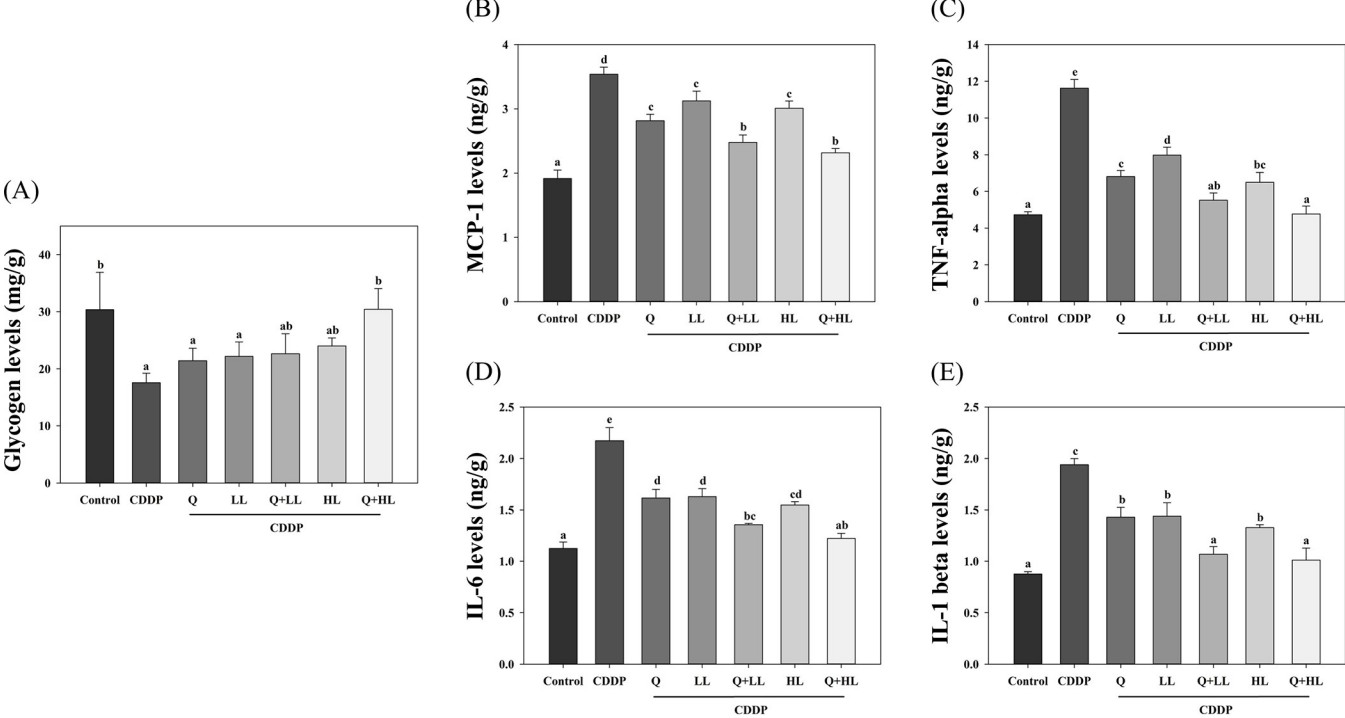

**Fig 8.** The individual and combined effect of quercetin (Q) and low dose (LL) or high dose (HL) of leucine on the levels of glycogen (A) and proinflammatory mediators, MCP-1 (B), TNF-alpha (C), IL-6 (D), and IL-1β (E) in triceps muscle in BALB/c mice exposed to cisplatin (CDDP). Values (mean ± SD) not sharing common letters are significantly different (one-way ANOVA, $p < 0.05$).

## Discussion

At present, there are many types of treatments available for patients with cancer, such as surgery, radiation therapy, immunotherapy, and targeted therapy [1]. CDDP, the first metal-based chemotherapeutic drug, still is one of the most common and irreplaceable chemotherapies [21], although it leads to lots of adverse effects including muscle, fat, and body weight loss, poor appetite, and CFR [4]. It has been shown that cancer patients who lost less weight after chemoradiation have better outcomes [22]. Therefore, the prevention of muscle, fat, and body weight loss is a crucial criterion for cancer patients undergoing chemotherapy. Our previous study has shown that quercetin, a flavonoid possessing antioxidant and anti-inflammatory effects, has moderately beneficial effects on TSA-induced muscle loss [8] and CDDP-induced fat loss [9]. The main objectives of the present study were to investigate the combined effects of quercetin and leucine, as well as the possible underlying mechanisms against CDDP-induced muscle atrophy and CRF; therefore, we first used CDDP-exposed BALB/c mice to investigate the issue. Our results demonstrated that the combination of quercetin and leucine (LL or HL) additively, holistically increased body, fat, and muscle weight and synergistically increased MGS in CDDP-exposed mice. The combined treatments also tended to enhance the locomotor activity compared with Q, LL and HL alone, although the combined effects were not synergistic. Furthermore, our results consistently showed that the combination of Q and HL synergistically increased muscle fiber size and MyHC expression in muscle tissues compared with the individual treatment (S5 and S6 Tables in S1 File). However, it is worth noting that the dose effects of leucine alone on those parameters mentioned above did not show significance.

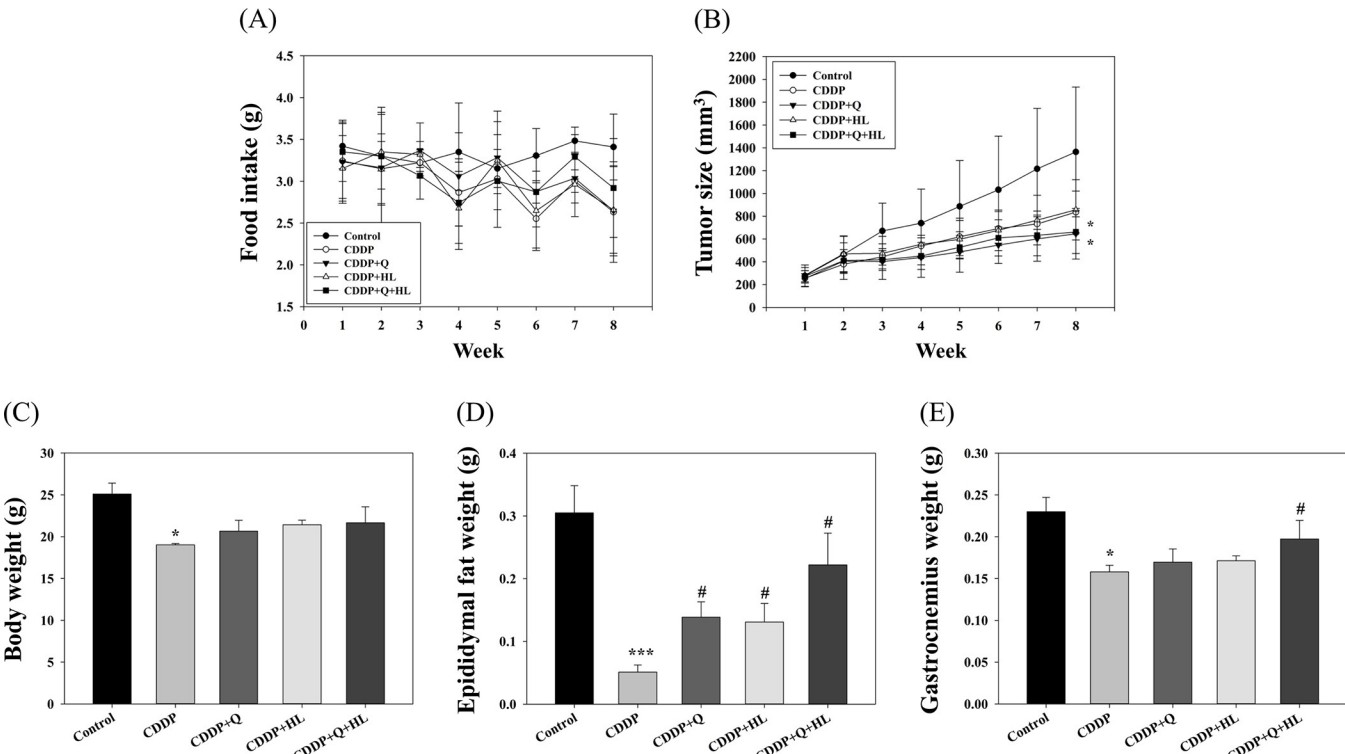

**Fig 9.** The individual and combined effect of quercetin (Q) and high dose of leucine (HL) on food intake (A), tumor size (B) as well as body (C), epididymal fat (D), and gastrocnemius (E) weight in tumor-bearing nude mice exposed to cisplatin (CDDP). CDDP was given by intraperitoneal injection (5 mg/kg B.W./ week). Values are presented as the mean ± SD. * denote a significant difference between the CDDP group and the control group; * $p < 0.05$ and *** $p < 0.001$; and # is significantly different between the supplemented group and the CDDP group; # $p < 0.05$ (Student's t-test).

Our results indicated that the prevention of the combined treatments in CDDP-induced weakness or CRF, which is reflected in increased MGS and locomotor activity, was associated with their effects on decreasing muscle wasting. It is well known that muscle wasting causes weakness and increases mortality and mobility in cancer patients. However, the association between muscle wasting and CRF in cancer patients is inconsistent [23]. The mechanisms underlying CRF in cancer patients are more complex than animal models and may include the regulation of the hypothalamic-pituitary-adrenal (HPA) axis [2]. Studies have demonstrated that CDDP causes muscle wasting *in vitro* and *in vivo* through activation of the Akt/FoxO1/ MuRF1/Atrogin-1 signaling pathway [24,25]. Our studies demonstrated the mechanism by which the combination of quercetin and leucine additively decreased CDDP-induced muscle wasting was associated with the downregulation of the signaling pathway and thus hampers muscle protein degradation. Quercetin and leucine alone have been shown to downregulate the FoxO1/MuRF1/Atrogin-1 pathway and decrease muscle wasting in trichostatin-treated tumor-bearing mice and in the C2C12 muscle cell model, respectively [8,26]. Our findings were in agreement with those studies and showed that the combined supplements consistently enhanced the suppression effect on the protein expression of MuRF1 and Atrogin-1. This data partly explained the increasing effect of the combined supplements on fiber size and MyHC expression. We only determined the protein expression of type IIa MyHC (a fast-twitch myofiber), but not other isoforms which include a slow-twitch myofiber, MyHC-I, and the other two fast-twitch myofibers, MyHC-IIx and MyHC-IIb. A recent study [27] has demonstrated that platinum-based anticancer drugs including cisplatin significantly reduce all the protein

levels of MyHC-I, MyHC-IIa, and MyHC-IIb, although the drugs only significantly decrease the gene expression of MyHC-IIa. The authors suggest that this is due to the degradation pathway of muscle proteins being the main mechanism contributing to CDDP-induced muscle atrophy. Because our study showed that the combination of quercetin and leucine enhanced the suppression effect on the activation of the Akt/FoxO1/MuRF1/Atrogin-1 pathway, we, therefore, speculated that the combined treatments could also increase MyHC-I and MyHC-IIb in muscles. However, further studies are needed to address this possibility.

It has been shown that leucine can promote muscle protein synthesis by activating the mTOR signaling pathway [14]. Our results also showed that leucine alone or in combination with quercetin upregulated the mTOR/p70 S6K/4E-BP1 pathway, indicating the contribution of the pathway in the protective effects of treatments on CDDP-induced muscle wasting. However, the enhanced effects of Q were only observed in the phosphorylation of 4E-BP1 rather than mTOR and p70 S6K, indicating the possibility that there were other molecules involved in the upregulation of 4E-BP1 phosphorylation. A study by Batool and his coworkers [28] also suggests that the phosphorylation dynamics of 4E-BP1 and p70 S6K1 are independently regulated. There are studies that show quercetin exerts an anticancer effect (breast cancer and pancreatic cancer) by inhibiting mTOR activation [29,30]. Quercetin also attenuates diabetic neuropathic pain by inhibiting the mTOR signaling pathway in db/db mice [31]. Our data showed that quercetin did not affect mTOR phosphorylation; this explains why the combined treatment did not enhance the phosphorylation of mTOR and p70 S6K. The difference between our study and others may be due to different types of cells (cancer cells vs. normal cells) or different pathological conditions. Because activated (phosphorylated) p70 S6K or 4E-BP1 alone can lead to mRNA translation and protein synthesis [32], results of the present study suggested that quercetin also played a crucial role in increasing muscle protein synthesis in an mTOR-independent way.

E2F-1 is a transcription factor that plays an important role in the control of cell cycle progression. E2F-1 also regulates cell growth by upregulating the expression of protein synthesis-associated genes, which is found to be essential for the induction of hypertrophy in C2C12 myoblasts without cell division [33]. In addition, the study by Real et al [16] shows that E2F-1 regulates cellular growth by activating mTOR signaling. E2F-1 disassociates from RB/E2F-1 complex and exerts its transcriptional activity, caused by the phosphorylation of RB by the Cyclin D/CDK4 complex [33,34]. Using next-generation sequencing, our preliminary study found that quercetin upregulated the mRNA expression of E2F-1 and its targets in normal human lung fibroblasts (IMR-90 cells) exposed to CDDP (S1 Fig). The present study showed that quercetin alone or in combination with leucine upregulated the Cyclin D/CDK4/RB/E2F-1 signaling pathway, suggesting that the activation of the signaling pathway, and then the promoting of cell cycle progression or cell growth may also contribute to the benefits of various supplements in muscle tissues in mice exposed to CDDP. The expression of PCNA, a cell growth marker [35], supported the speculation, although the expression of PCNA was similar among the groups administered various supplements. In fact, we also determined the expression of Ki-67, which is only present during active phases of the cell cycle and is often used as a marker for cell proliferation [36,37] in quadriceps muscle by IHC staining. The results were consistent with the findings of the E2F-1 signaling pathway in gastrocnemius muscles, that is, the combined treatment, especially Q+HL, markedly increased the numbers of Ki-67 positive cells (S2 Fig). Our study provides a novel insight into the possible mechanisms by which quercetin alone or in combination with leucine provides protective effects in mice exposed to CDDP. However, further studies including determining the changes in the number of morphological myofibers in gastrocnemius muscle tissues are warranted to investigate the precise roles of E2F-1 signaling in muscle tissues in mice exposed to CDDP.

Besides muscle mass, it has been suggested that glycogen levels in skeletal muscle also contribute to muscle strength [20]. Researchers have shown that decreasing glycogen levels lead to a reduction in $Ca^{2+}$ release from the sarcoplasmic reticulum which resulted in muscle fatigue [20,38]. In agreement with the above statement, the present study also showed decreased glycogen levels in CDDP-exposed mice. Treatment with quercetin and leucine in combination tended to increase glycogen levels better than individual treatment and were positively corroborated with improved MGS and locomotor activity. Chen and her colleagues [39] showed that dietary quercetin supplementation promoted anti-fatigue activity and enhanced muscle function via increasing antioxidant capacity and glycogen storage in mice. The precise mechanism underlying the effects of quercetin and leucine alone or combined on glycogen levels in the triceps muscle remains unclear. We speculated it might be associated with increased food intake because our data showed that LL or HL alone or in combination with Q significantly increased food intake in mice exposed to CDDP.

Several studies have demonstrated that chemotherapy-induced cachexia or CRF is associated with an increase in proinflammatory cytokines, such as TNF-α, IL-6, and IL-1β [40,41]. CDDP alters protein metabolism by increasing oxidative stress and proinflammatory cytokine secretion, which in turn increases skeletal muscle catabolism and BCAA oxidation [42]. A review study points out that proinflammatory cytokines upregulate the ubiquitin-proteasome pathway (UPP), that is the MuRF1/atrogin-1 associated pathway, by inducing NF-κB [43]. Thus, our data suggested that the combination of quercetin and leucine downregulated UPP activation through a mechanism associated with decreasing proinflammatory cytokine levels. In addition, the combined treatments also decreased the levels of MCP-1, a chemokine that has an important role in the process of inflammation by attracting or enhancing the expression of other inflammatory mediators/cells and is involved in various diseases including CRF [2,44].

By comparing the observed and respective effects of the combined treatment, we determined whether the combined effects of quercetin and leucine on various parameters were synergistic or not. Although the combined treatments, especially Q+HL, synergistically increased fiber CSA and the protein level of MyHC, the combined effects on signaling molecules present in muscle tissues (S6-S8 Tables in S1 File) were only additive or less than additive (antagonistic), indicating that each of these signaling pathways could only partly explain the effects of combined treatments on preventing muscle wasting. Similar situations were also found in the comparison of the combined effects on the level of glycogen and proinflammatory cytokines, suggesting the regulation of these parameters by the combined treatments also only partly contributed to their effects on MGS.

Using tumor-bearing nude mice, the present study demonstrated that quercetin (200 mg/kg B.W./day) and HL alone or combined neither compromised nor enhanced the anticancer effect of CDDP at 5 mg/kg. Our previous study showed that quercetin alone given by a diet containing 1% quercetin (about 1200 mg/kg B.W./day) or by intraperitoneal injection at a dose of 10 mg/kg, 3 times a week significantly enhanced the anticancer effect of CDDP at 2 mg/kg [9]. The differences between the two studies on the anticancer effect of CDDP may be due to the different doses of quercetin and CDDP. However, we still observed that quercetin alone significantly attenuated CDDP-induced fat loss in the present study. Consistent with what we observed in BALB/c mice, the combined effects of Q and HL also tended to be better than the individual effect on increasing the weight of epididymal fat and gastrocnemius muscle in tumor-bearing nude mice, indicating the potential that using the combined treatments prevents CDDP-induced fat loss and improves muscle mass.

There are some limitations in the present study. First, we used different muscles to perform different analyses: quadriceps to determine CSA; gastrocnemius to determine MyHC and

signaling molecules; and triceps to determine glycogen and cytokine levels. This situation led to indirect evidence about the molecular mechanisms underlying the protective effects of combined treatments on CDDP-induced muscle wasting. Because the quadriceps muscles were stored in 10% formalin for histological analysis, we used gastrocnemius muscles to perform western blotting to determine signaling molecules. In addition, as we have mentioned above, we analyzed the glycogen and cytokine levels of the triceps muscle, because this muscle is the muscle located in the forelimb, which was used to measure the MGS. However, the quadriceps, gastrocnemius, and triceps are all fast muscles; they are fast-twitch myofiber-dominant. Our study showed that CDDP similarly decreased the weight of all three muscles by about 30% (Table 1). Both the quadriceps and gastrocnemius are commonly used to determine CSA, MyHC, and the signaling molecules associated with protein degradation and synthesis. CDDP has similar effects on these two muscle tissues [27,45]. Therefore, our study still provided some useful evidence. Second, we did not determine the parameters mentioned above in the soleus muscles (the only slow-twitch myofiber-dominant muscle determined in the present study), although the combined treatment of Q and HL appeared to have a better and similar effect on recovering the weight of the quadriceps and soleus by 29% and 33%, respectively, compared to other muscles. The precise reasons for Q+HL in recovering the soleus muscle remain unclear. However, the study by Haegens et al. [46] shows that leucine-induced up-regulation of slow MyHC seems to be better than fast MyHC. The authors found that the mRNA expression of MyHC-7 (the gene encodes MyHC-I) is in an mTOR-independent manner while MyHC-4 (the gene encodes MyHC-IIb) is in an mTOR-dependent manner. Our data also showed that quercetin increased muscle protein synthesis in an mTOR-independent manner. This may explain why the combination of Q and HL had a better recovery effect on the soleus. However, more studies are needed to confirm the mechanisms underlying the effect of Q+HL on the soleus. Third, as we have mentioned above, the evidence for cell proliferation of muscle tissues also was indirect. Further studies are warranted, including cellular studies, to investigate the precise role of upregulation of the E2F-1 signaling pathway in muscle tissues by the combined treatments.

## Conclusion

The present study demonstrates that the combination of quercetin and leucine (Q+HL or Q+LL) was more effective than the individual supplement in improving CDDP-induced weakness and CRF by decreasing muscle wasting and increasing glycogen levels in muscle tissues without compromising the anticancer effect of CDDP in mice. Multiple mechanisms may contribute to the attenuation of muscle atrophy, including the downregulation of the FoxO1 signaling pathway as well as the upregulation of the mTOR and E2F-1 signaling pathways. Further studies including cellular studies are needed to support the above results.

## Supporting information

**S1 Fig. Gene set enrichment analysis (GSEA) showed an enrichment of E2F target genes in IMR-90 cells (ATCC CCL-186, human lung fibroblasts) exposed to cisplatin + quercetin.** The cells were incubated in Eagle's Minimum Essential Medium supplemented with 10% (v/v) fetal bovine serum and 1% penicillin-streptomycin at 37 ˚C in a humidified atmosphere of 5% $CO_2$. After co-incubation with cisplatin (1 μM) and quercetin (5 μM) for 48 hours, the total RNA of cells was collected for the next generation sequencing (NGS) and GSEA.
(TIFF)

**S2 Fig. The individual and combined effect of quercetin (Q) and low dose (LL) or high dose (HL) of leucine on Ki-67 protein expression in the quadriceps muscle in BALB/c mice exposed to cisplatin (CDDP).** Immunohistochemical staining was performed by the Ultra-View Universal DAB Detection Kit (Roche, Switzerland) and Ki-67 antibody (cat #: 12202, Cell Signaling Technology) and the sections were examined using the Tissue Cytometer (TissueGnostics, Vienna, Australia; magnification, x200). The nuclei were stained in blue with Hematoxylin and Ki-67-positive cells were stained brown. Bar in the picture is 20 μm and the area framed by the rectangles represents Ki-67-positive cells.
(TIFF)

**S3 Fig. The original gel image underlying Fig 5A blot results.**
(JPG)

**S4 Fig. The original gel image underlying Fig 5B blot results.**
(JPG)

**S5 Fig. The original gel image underlying Fig 5C blot results.**
(JPG)

**S6 Fig. The original gel image underlying Fig 6 blot results).**
(JPG)

**S7 Fig. The original gel image underlying Fig 7A blot results.**
(JPG)

**S8 Fig. The original gel image underlying Fig 7B blot results.**
(JPG)

**S1 File. S1-S10 Tables.** Comparison between observed inhibition and expected inhibition of quercetin (Q) in combination with low dose (LL) or high dose of leucine (HL) on the levels of parameters determined in BALB/c mice (S1-S9 Tables) or tumor-bearing nude mice (S10 Table) exposed to cisplatin.
(DOCX)

## Author Contributions

**Conceptualization:** Shu-Lan Yeh.

**Formal analysis:** Te-Hsing Hsu, Jiunn-Wang Liao.

**Funding acquisition:** Shu-Lan Yeh.

**Investigation:** Te-Hsing Hsu, Ting-Jian Wu, Yu-An Tai.

**Methodology:** Te-Hsing Hsu, Ting-Jian Wu, Yu-An Tai, Shu-Lan Yeh.

**Resources:** Chin-Shiu Huang, Jiunn-Wang Liao.

**Supervision:** Shu-Lan Yeh.

**Validation:** Te-Hsing Hsu, Shu-Lan Yeh.

**Writing – original draft:** Te-Hsing Hsu.

**Writing – review & editing:** Shu-Lan Yeh.

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
