## [Decision Letter · Decision Letter 0]

3 Jul 2023

PONE-D-23-17913The combination of quercetin and leucine synergistically improves grip strength by attenuating muscle atrophy by multiple mechanisms in mice exposed to cisplatinPLOS ONE

Dear Dr. Yeh,

Thank you for submitting your manuscript to PLOS ONE. After careful consideration, we feel that it has merit but does not fully meet PLOS ONE’s publication criteria as it currently stands. Therefore, we invite you to submit a revised version of the manuscript that addresses the points raised during the review process.

We look forward to receiving your revised manuscript.

Kind regards,

Hiroshi Kaji

Academic Editor

PLOS ONE

Journal Requirements:

   "This research was supported by grants (MOST 108-2320-B-040-014-MY2) from the National Science and Technology Council, Republic of China."

   "This research was supported by grants (MOST 108-2320-B-040-014-MY2) from the National Science and Technology Council, Republic of China."

   "This research was supported by grants (MOST 108-2320-B-040-014-MY2) from the National Science and Technology Council, Republic of China."

7. PLOS ONE now requires that authors provide the original uncropped and unadjusted images underlying all blot or gel results reported in a submission’s figures or Supporting Information files. This policy and the journal’s other requirements for blot/gel reporting and figure preparation are described in detail at https://journals.plos.org/plosone/s/figures#loc-blot-and-gel-reporting-requirements and https://journals.plos.org/plosone/s/figures#loc-preparing-figures-from-image-files. When you submit your revised manuscript, please ensure that your figures adhere fully to these guidelines and provide the original underlying images for all blot or gel data reported in your submission. See the following link for instructions on providing the original image data: https://journals.plos.org/plosone/s/figures#loc-original-images-for-blots-and-gels. 

Additional Editor Comments:

The reviewers raised several concerns in this paper, which should be  adequately addressed or some rebuttal to each comment . Especially, statistical analyses should be adequately performed with the help of statistician. 

Reviewers' comments:

Reviewer's Responses to Questions

**Comments to the Author**

1. Is the manuscript technically sound, and do the data support the conclusions?

Reviewer #1: Partly

Reviewer #2: Yes

2. Has the statistical analysis been performed appropriately and rigorously? 

Reviewer #1: No

Reviewer #2: Yes

3. Have the authors made all data underlying the findings in their manuscript fully available?

Reviewer #1: No

Reviewer #2: Yes

4. Is the manuscript presented in an intelligible fashion and written in standard English?

Reviewer #1: No

Reviewer #2: Yes

5. Review Comments to the Author

Reviewer #1: The authors examine the combined effects of quercetin (Q) and leucine (L) on cisplatin (CDDP)-induced muscle atrophy and cancer-related fatigue (CRF) in this study. Synergistic effects of Q and L were demonstrated in grip strength and FCSA. However, there is no two-way ANOVA interaction in the analysis of molecular mechanisms such as protein degradation and synthesis systems, and the association with the conclusions is not sufficient. Appropriate statistical processing should be performed, the results should be reinterpreted, and the description of the results and discussion should be revised.

1. The authors perform group comparisons even when there is no interaction in a two-way ANOVA. Group comparisons can only be performed when there are interactions in two-way ANOVA. The p-values for Q, L, and all interactions by two-way ANOVA should be clarified.

2. The authors use Duncan's multiple range test for the posterior test. This test does not account for multiplicity issues and should not be used. Post hoc tests such as the Turkey HSD and Bonferroni, which are commonly used in a wide range of fields, should be used.

3. The authors examine the effects of Q and L on cell cycle and proliferation analysis. Changes in the number of morphological myofibers are not shown, and the consistency with the results of the analysis of molecular mechanisms is unclear.

4. The interval of the grip strength measurement is not described.

5. The photographic area and number of photographs used to measure FCSA are not described.

6. There are typos in Lines 238, 268, and 346.

7. Quadriceps is a complex muscle, but it is not indicated which region was analyzed.

8. The details (model number) of the antibodies used are not given.

9. In Fig. 4, the muscles analyzed for FCSA and MyHC are different, but the reason is not indicated.

10. MyHC isoforms are not shown, and it is not possible to consider whether they indicate a transition in myofiber type or a change in myofiber size.

11. In Fig. 4, CSA of muscle fibers cannot be observed.

12. Molecular weight not shown in WB images.

13. The description of the statistical treatment method used for each Fig is unclear (e.g., Fig. 9A).

14. Information on the intake of tumor-bearing nude mice is not shown.

15. The authors did not visualize the data in all graphs, all data should be visualized graphically.

16. What statistical results do the terms "synergistically" and "additively" reflect?

17. Limitations are not described.

Reviewer #2: In this study, Hsu et al . studied the effects of the combination of quercetin and leucine on cisplatin-induced muscle atrophy and cancer-related fatigue in mice and the mechanisms by which the combination of quercetin and leucine on cisplatin-induced muscle atrophy and cancer-related fatigue. The authors revealed that the combination of quercetin and leucine synergistically or additively blunted cisplatin-induced decreased in body weight, grip strength, locomotor activity, fat and muscle weights, muscle fiber size and MyHC levels in muscle tissues. The combined treatments decreased atrogin-1, MuRF1, and proinflammatory cytokine levels. Moreover, the combined treatments increased mTOR and E2F-1 levels as well as glycogen levels in muscle tissues. Finally, the authors showed that the combined treatments blunted canver-related fatigue without influence of anticancer effects of cisplatin in mice. However, there are several issues with the manuscript. The details are attached below.

Major points

1. The combined treatments blunted the decrease in CSA, phosphorylation of Akt and FoxO1, and mTOR signaling of the fast-twitch myofiber-dominant gastrocnemius muscles. Does the combined treatments blunt the decrease in CSA and Akt/mTOR or FoxO1 signaling in the slow-twitch myofiber-dominant soleus muscles?

2. Fig 4D: Which types of MyHC were recognized by the anti-MyHC antibodies? Does the combined treatments increased MHC-I, MHC-IIa and MHC-IIb levels?

3. Although the gastrocnemius muscles were used in Figs 4, 5, 6, 7, the triceps were used for analyses of glycogen and cytokine levels in Fig 8. In Fig 3, quadriceps were used for analyses of CSA. Please provide a rationale for the used muscle tissues.

4. Table 1: The combined treatments to cisplatin-treated mice completely recovered tissue weight of soleus muscles, but partially tissue weights of triceps, quadriceps, gastrocnemius, and tibialis anterior muscles. Please add the discussion regarding these results.

6. PLOS authors have the option to publish the peer review history of their article (what does this mean?). If published, this will include your full peer review and any attached files.

Reviewer #1: No

Reviewer #2: No

---

## [Author Response · Author response to Decision Letter 0]

14 Aug 2023

Reviewer #1:

We thank the reviewer for the constructive comments. We have now corrected the statistical methods including using ANOVA with Tukey post hoc test to perform group comparisons. The results and discussion have been revised carefully with the new statistical results according to the suggestion of the reviewer.

The following are our point-by-point responses.

1.The authors perform group comparisons even when there is no interaction in a two-way ANOVA. Group comparisons can only be performed when there are interactions in two-way ANOVA. The p-values for Q, L, and all interactions by two-way ANOVA should be clarified.

[ANS]: In fact, we used one-way ANOVA to perform group comparisons. The two-way ANOVA was only performed to determine the interaction (or synergistic effect) between Q and HL or Q and LL. However, the statistical method may be confusing and not proper. We have now determined whether the combined preventive effects of quercetin and leucine on CDDP-induced damages were synergistic or not by comparing the differences between the observed inhibitions and expected inhibitions of combined treatments using one-sample t-test as described previously (Yeh et al., 2009). When the observed inhibition of the combined treatment for one parameter was significantly better than its expected inhibition, a synergistic effect was evident; when the difference between the observed and expected inhibitions was not significant, an additive effect was indicated. The details have been described in the Methods. We conducted the analysis for all parameters we determined including molecular mechanisms such as protein degradation and synthesis systems. We also added some discussion about the results (the 7th paragraph). The data are shown in Supporting Tables. 

2. The authors use Duncan's multiple range test for the posterior test. This test does not account for multiplicity issues and should not be used. Post hoc tests such as the Tukey HSD and Bonferroni, which are commonly used in a wide range of fields, should be used.

[ANS]: We have now used Tukey HSD to perform posterior tests.

3. The authors examine the effects of Q and L on cell cycle and proliferation analysis. Changes in the number of morphological myofibers are not shown, and the consistency with the results of the analysis of molecular mechanisms is unclear.

[ANS]: We did not determine the changes in the number of morphological myofibers but have determined the expression of PCNA. In addition, we also determined the expression of Ki-67, which is only present during active phases of the cell cycle and is often used as a marker for cell proliferation (Scholzen and Gerdes, 2000; Chinzei et al., 2015) in quadriceps muscle by IHC staining. The results were consistent with the findings of cell cycle and proliferation analysis in gastrocnemius muscles, that is, the combined treatment, especially Q+HL, markedly increased the numbers of Ki-67 positive cells. We have added the statement to the Discussion (the 4th paragraph) and the data to S2 Fig). However, the evidence may be indirect and limited because we used different parts of muscle tissues to conduct these studies. In addition, we did not conduct a cellular study to confirm these findings. We have added this limitation to the Discussion.

4. The interval of the grip strength measurement is not described.

[ANS]: The interval between each grip strength test was one minute. We have added this information to the Methods.

5. The photographic area and number of photographs used to measure FCSA are not described.

[ANS]: We have now added the information as follows: “... quadriceps samples were examined using Tissue Cytometer (TissueGnostics, Vienna, Australia; magnification, x200). Myofiber CSAs of the rectus femoris region were calculated using TissueFAXS Viewer software (TissueGnostics, Vienna, Australia), and a minimum of 8 random images and 200 sets (25 sets/image) of data were acquired per group.

6. There are typos in Lines 238, 268, and 346.

[ANS]: We thank the reviewer for pointing out the errors. We have corrected the errors in the revised version.

7. Quadriceps is a complex muscle, but it is not indicated which region was analyzed.

[ANS]: We used the rectus femoris region of the quadriceps to determine CSA. We have added the information to the Methods. 

8. The details (model number) of the antibodies used are not given.

[ANS]: We have added the cat. number of antibodies to the Methods.

9. In Fig. 4, the muscles analyzed for FCSA and MyHC are different, but the reason is not indicated.

[ANS]: Because the quadriceps muscles were stored in 10% formalin for histological analysis, we used gastrocnemius muscles to conduct western blotting for determining the protein expression of MyHC as well as the molecules which are associated with signaling pathways. Quadriceps and gastrocnemius are all fast muscles. Our study showed that CDDP similarly decreased all of these three muscles' weight by about 30% (Table 1). Both quadriceps and gastrocnemius are commonly used to determine CSA and the expression of molecules associated with protein degradation and synthesis signaling pathways. CDDP has similar effects on the parameters of these two muscle tissues (Sato et al., 2003; Chi et al., 2022).

We have now added this information to the Discussion (the last paragraph). In addition, we also moved the figure of MyHC expression to Figure 5 to avoid confusion.

10. MyHC isoforms are not shown, and it is not possible to consider whether they indicate a transition in myofiber type or a change in myofiber size.

[ANS]: The MyHC we determined is type IIa. We have added this information to the Methods. Both quadriceps and gastrocnemius are fast-twitch myofiber-dominant muscles. Thus, type IIa MyHC could be a protein content index to confirm the change of myofiber size.

11. In Fig. 4, CSA of muscle fibers cannot be observed.

[ANS]: Fig. 4B was the result of CSA of muscle fibers. We have now improved the figure quality.

12. Molecular weight not shown in WB images.

[ANS]: As suggested, we have added the molecular weight to the WB images.

13. The description of the statistical treatment method used for each Fig is unclear (e.g., Fig. 9A).

[ANS]: We have added the statistical method used for each figure.

14. Information on the intake of tumor-bearing nude mice is not shown.

[ANS]: As suggested, we have added the intake of tumor-bearing mice (Fig 9A) to the results. The differences among groups were not significant.

15. The authors did not visualize the data in all graphs, all data should be visualized graphically.

[ANS]: We have improved the quality of all graphs.

16. What statistical results do the terms "synergistically" and "additively" reflect?

[ANS]: As we have mentioned above, we have now determined that the combined preventive effects of quercetin and leucine on CDDP-induced damages were synergistic or additive by comparing the differences between the observed inhibitions and expected inhibitions of combined treatments using one-sample t-test as described previously (Yeh et al., 2009). The observed inhibition of the combined treatment for one parameter was significantly better than its expected inhibition, indicating a synergistic effect; while the difference was not significant, indicating an additive effect. The details have been described in the Methods. 

17. Limitations are not described.

[ANS]: As suggested, we have now described the limitations at the end of the Discussion as follows:

There are some limitations in the present study. First, we used different muscles to perform different analyses: quadriceps to determine CSA; gastrocnemius to determine MyHC and signaling molecules; and triceps to determine glycogen and cytokine levels. This situation led to indirect evidence about the molecular mechanisms underlying the protective effects of combined treatments on CDDP-induced muscle wasting . Because the quadriceps muscles were stored in 10% formalin for histological analysis, we used gastrocnemius muscles to perform western blotting to determine signaling molecules. In addition, as we have mentioned above, we analyzed the glycogen and cytokine levels of the triceps muscle, because this muscle is the muscle located in the forelimb, which was used to measure the MGS. However, the quadriceps, gastrocnemius, and triceps are all fast muscles; they are fast-twitch myofiber-dominant. Our study showed that CDDP similarly decreased the weight of all three muscles by about 30% (Table 1). Both the quadriceps and gastrocnemius are commonly used to determine CSA, MyHC, and the signaling molecules associated with protein degradation and synthesis. CDDP has similar effects on these parameters of these two muscle tissues [27,45]. Therefore, our study still provided some useful evidence. Second, we did not determine the parameters mentioned above in the soleus muscles (the only slow-twitch myofiber-dominant muscle determined in the present study), although the combined treatment of Q and HL appeared to have a better and similar effect on recovering the weight of the quadriceps and soleus by 29% and 33%, respectively, compared to other muscles. The precise reasons for Q+HL in recovering the soleus muscle remain unclear. However, the study by Haegens et al. [46] shows that leucine-induced up-regulation of slow MyHC seems to be better than fast MyHC. The authors found that the mRNA expression of MyHC-7 (the gene encodes MyHC-I) is in an mTOR-independent manner while MyHC-4 (the gene encodes MyHC-IIb) is in an mTOR-dependent manner. Our data also showed that quercetin increased muscle protein synthesis in an mTOR-independent manner. This may explain why the combination of Q and HL had a better recovery effect on the soleus. However, more studies are needed to confirm the mechanisms underlying the effect of Q+HL on the soleus. Third, as we have mentioned above, the evidence for cell proliferation of muscle tissues also was indirect. Further studies are warranted, including cellular studies, to investigate the precise role of upregulation of the E2F-1 signaling pathway in muscle tissues by the combined treatments.”

References

Chi MY, Zhang H, Wang YX, Sun XP, Yang QJ, Guo C. Silibinin Alleviates Muscle Atrophy Caused by Oxidative Stress Induced by Cisplatin through ERK/FoxO and JNK/FoxO Pathways. Oxid Med Cell Longev. 2022; 2022: 5694223.

Chinzei N, Hayashi S, Ueha T, Fujishiro T, Kanzaki N, Hashimoto S, Sakata S, Kihara S, Haneda M, Sakai Y, Kuroda R, Kurosaka M. P21 deficiency delays regeneration of skeletal muscular tissue. PLoS One. 2015; 10: e0125765.

Sato K, Miyauchi Y, Xu X, Kon R, Ikarashi N, Chiba Y, Hosoe T, Sakai H. Platinum-based anticancer drugs-induced downregulation of myosin heavy chain isoforms in skeletal muscle of mouse. J Pharmacol Sci. 2023; 152: 167-177.

Scholzen T, Gerdes J. The Ki-67 protein: from the known and the unknown. J Cell Physiol. 2000; 182: 311–322.

Yeh SL, Wang HM, Chen PY, Wu TC. Interactions of beta-carotene and flavonoids on the secretion of pro-inflammatory mediators in an in vitro system. Chem Biol Interact. 2009; 179: 386–393.

Reviewer #2: 

We thank the reviewer for the constructive comments. The following are our point-by-point responses.

1. The combined treatments blunted the decrease in CSA, phosphorylation of Akt and FoxO1, and mTOR signaling of the fast-twitch myofiber-dominant gastrocnemius muscles. Does the combined treatments blunt the decrease in CSA and Akt/mTOR or FoxO1 signaling in the slow-twitch myofiber-dominant soleus muscles?

[ANS]: We did not determine CSA and the expression of Akt/mTOR or FoxO1 signaling in soleus muscles (the slow-twitch myofiber-dominant muscle). The study by Sato et al. (2023) has demonstrated that platinum-based anticancer drugs including cisplatin significantly reduce all the protein levels of MyHCs, including MyHC-I (a slow-twitch myofiber) by the degradation pathway (more details have been described in the answer of the next question). Our study showed that the combined treatments increased the weight of various muscles including soleus muscles compared with the CDDP alone group. Thus, we speculated the combined treatments could regulate those parameters in soleus muscles, too. However, further studies are needed to confirm this speculation and whether they regulated mTOR signaling in the soleus. We have added the information to the Discussion (the last paragraph).

2. Fig 4D: Which types of MyHC were recognized by the anti-MyHC antibodies? Does the combined treatments increase MHC-I, MHC-IIa and MHC-IIb levels?

[ANS]: The MyHC we determined is type IIa. We have added this information to the Methods.

A recent study (Sato et al., 2023) has demonstrated that platinum-based anticancer drugs including cisplatin significantly reduce all the protein levels of MyHC-I, MyHC-IIa and MyHC-IIb, although the anticancer drugs only significantly decrease the gene expression of MyHC-IIa. The authors suggest that this is due to the degradation pathway of muscle proteins being the main mechanism contributing to cisplatin-induced muscle atrophy. Because our study showed that the combined treatments could further decrease the activation of the Akt/FoxO1/MuRF1/atrogin-1 associated protein degradation pathway, we, therefore, speculated that the combined treatments could also increase MyHC-I and MyHC-IIb in muscles. However, more studies are needed to address this possibility. We have now added this information to the Discussion (the 2nd paragraph).

3. Although the gastrocnemius muscles were used in Figs 4, 5, 6, 7, the triceps were used for analyses of glycogen and cytokine levels in Fig 8. In Fig 3, quadriceps were used for analyses of CSA. Please provide a rationale for the used muscle tissues.

[ANS]: Quadriceps, gastrocnemius, and triceps are all fast muscles. Our study showed that CDDP similarly decreased all of these three muscles' weight by about 30% (Table 1). Both quadriceps and gastrocnemius are commonly used to determine CSA, the regulation of proteins associated with protein degradation and synthesis signaling pathways. CDDP has similar effects on those parameters of these two muscle tissues (Sato et al., 2003; Chi et al., 2022). Because the quadriceps muscles were stored in 10% formalin for histological analysis, we used gastrocnemius muscles to conduct western blotting to determine the protein expression of MyHC as well as the molecules which are associated with FoxO1 signaling, mTOR signaling pathway, and E2F-1 signaling pathway. In addition, because the forelimb of the mice was used to measure the MGS, we, therefore, analyzed the glycogen and cytokine levels of the triceps muscle, which is the muscle located in the forelimb. We have now added this rationale to the Discussion (the last paragraph).

4. Table 1: The combined treatments to cisplatin-treated mice completely recovered tissue weight of soleus muscles, but partially tissue weights of triceps, quadriceps, gastrocnemius, and tibialis anterior muscles. Please add the discussion regarding these results.

[ANS]: Based on the suggestion of reviewer #1, we have now used ANOVA with Tukey HSD (instead of Duncan's multiple range) to conduct the statistical analysis for the data in Table 1. The results showed that the combined treatments had better effects on recovering the weight of the quadriceps and soleus (the only slow muscle that we determined) by 29% and 33%, respectively, compared to other muscle tissues. The precise reasons for Q+HL in recovering the soleus muscle remain unclear. The study by Haegens et al. (2012) shows that leucine-induced up-regulation of slow MyHC seems to be better than fast MyHC. The authors found that the mRNA expression of MyHC-7 (the gene encodes MyHC-I) is in an mTOR-independent manner while MyHC-4 (the gene encodes MyHC-IIb) is in an mTOR-dependent manner. Our data also showed that quercetin increased muscle protein synthesis in an mTOR-independent manner. This may explain why the combination of Q and HL had a better recovery effect on the soleus. However, more studies are needed to confirm the mechanisms underlying the effect of Q+HL on the soleus. We have added the information to the Discussion (the last paragraph). 

Chi MY, Zhang H, Wang YX, Sun XP, Yang QJ, Guo C. Silibinin Alleviates Muscle Atrophy Caused by Oxidative Stress Induced by Cisplatin through ERK/FoxO and JNK/FoxO Pathways. Oxid Med Cell Longev. 2022; 2022: 5694223.

Haegens A, Schols AM, van Essen AL, van Loon LJ, Langen RC. Leucine induces myofibrillar protein accretion in cultured skeletal muscle through mTOR dependent and -independent control of myosin heavy chain mRNA levels. Mol Nutr Food Res. 2012; 56: 741-752.

Sato K, Miyauchi Y, Xu X, Kon R, Ikarashi N, Chiba Y, Hosoe T, Sakai H. Platinum-based anticancer drugs-induced downregulation of myosin heavy chain isoforms in skeletal muscle of mouse. J Pharmacol Sci. 2023; 152: 167-177.

---

## [Decision Letter · Decision Letter 1]

29 Aug 2023

The combination of quercetin and leucine synergistically improves grip strength by attenuating muscle atrophy by multiple mechanisms in mice exposed to cisplatin

PONE-D-23-17913R1

Dear Dr. Yeh,

We’re pleased to inform you that your manuscript has been judged scientifically suitable for publication and will be formally accepted for publication once it meets all outstanding technical requirements.

Kind regards,

Hiroshi Kaji

Academic Editor

PLOS ONE

Additional Editor Comments (optional):

Reviewers' comments:

Reviewer's Responses to Questions

**Comments to the Author**

1. If the authors have adequately addressed your comments raised in a previous round of review and you feel that this manuscript is now acceptable for publication, you may indicate that here to bypass the “Comments to the Author” section, enter your conflict of interest statement in the “Confidential to Editor” section, and submit your "Accept" recommendation.

Reviewer #1: All comments have been addressed

Reviewer #2: All comments have been addressed

2. Is the manuscript technically sound, and do the data support the conclusions?

Reviewer #1: Yes

Reviewer #2: Yes

3. Has the statistical analysis been performed appropriately and rigorously? 

Reviewer #1: Yes

Reviewer #2: Yes

4. Have the authors made all data underlying the findings in their manuscript fully available?

Reviewer #1: Yes

Reviewer #2: Yes

5. Is the manuscript presented in an intelligible fashion and written in standard English?

Reviewer #1: Yes

Reviewer #2: Yes

6. Review Comments to the Author

Reviewer #1: The revised manuscript is significantly improved in many respects. The methods and results are more clearly described, and the limitations of the study are more extensively described. The authors have addressed all of the previously raised concerns.

Reviewer #2: The revised manuscript brought additional information. The authors well addressed most of my concerns by revising the paper.

7. PLOS authors have the option to publish the peer review history of their article (what does this mean?). If published, this will include your full peer review and any attached files.

Reviewer #1: No

Reviewer #2: No

---

## [Editor Report · Acceptance letter]

4 Sep 2023

PONE-D-23-17913R1 

The combination of quercetin and leucine synergistically improves grip strength by attenuating muscle atrophy by multiple mechanisms in mice exposed to cisplatin 

Dear Dr. Yeh:

I'm pleased to inform you that your manuscript has been deemed suitable for publication in PLOS ONE. Congratulations! Your manuscript is now with our production department. 

Kind regards, 

on behalf of

Dr. Hiroshi Kaji 

Academic Editor

PLOS ONE